

# The analysis of H/V curve from different ellipticity retrieval technique for a single 3c-station recording.

Irfan Ullah[1], Renato Luiz Prado[1].

1.Departamento de Geofísica do Instituto de Astronomia, Geofísica e Ciências Atmosféricas

Rua do Matão, 1226 - Cidade Universitária São Paulo-SP - Brasil

*Correspondence to*: Irfan Ullah (syed_528@yahoo.com)

**Abstract.** In the last two decades or so the H/V (Horizontal-to-Vertical Spectral Ratio) technique remained very popular, and are extensively used for the site fundamental frequency estimation. H/V curve are also used with dispersion curve to jointly invert and retrieved the shear wave velocity of relatively deep soil deposit. Although a full theoretical explanation of H/V technique is not been presented yet, There are two main assumptions used generally that H/V curves can be explained by considering Rayleigh wave of noise wave field only while the other newly presented approach utilized the whole noise wave field known as diffuse field approach (DFA), However in case of Rayleigh wave approach for H/V, it is almost impossible to the remove the fraction of Love wave to the horizontal component of H/V. Here in this study we aim to test different approaches adopted for the removal of Love wave fraction from horizontal component for a borehole test site at University of Sao Paulo. The result from different approaches are compared with borehole ellipticity curve. The result shows that around the fundamental frequency of curve obtained in either way(DFA or ellipticity approach) is dominated by Rayleigh waves.

## 1. Introduction.

H/V (Horizontal-to-Vertical spectral ratio) is fast and quick way to get properties of a site for engineering interest, by the measurement of ambient noise wave field with a single 3-component sensor on the earth surface. The method is used for rapid estimation of fundamental resonance frequency ($f_0$) of a site where the maximum displacement amplification is expected in case of an earthquake. This technique is also used to retrieve the shear wave velocity of the geological structure in a joint inversion with dispersion curve (Scherbaum et al, 2003 Picozi et al, 2005 Hobiger et al, 2013). However, some controversies about the nature of ambient noise wave field and sources exist, which most of the time make the results of H/V curve questionable and hence debatable. Apart from the controversies exist in nature of ambient noise wave filed M. Mucciarelli et al (2001) described some problem regarding the





acquisition and processing of H/V spectral ratio. Due to extensive utilization of  H/V technique, a
commission was established to test the acquisition, processing and interpretation of this technique (site
effects assessment using ambient excitations) – the SESAME project (2001-2004), The guidelines
reports are published for acquisition, processing and interpretation of ambient noise wave field and have
addressed all the points raised and discussed by M. Mucciarelli et al (2001) in extensive details. The
H/V curve are modeled with different approaches and each modeling approach might have the effect on
the retrieved soil profile(Sánchez-Sesma et al, 2011, Lunedei & Malischewsky  2015). The acquisition
of the data is made with a three component sensor placed on the surface of ground which record the
seismic noise wave field. Fourier spectra of the recorded seismic noise  for all the three component (eat-
west , north-south and vertical) are made. The two horizontal component Fourier spectra are properly
averaged and then divided by the vertical Fourier spectra. This division of  averaged horizontal and
vertical component result a curve (H/V) as function of frequency. H/V curve usually result a peak
depending on the subsoil stratigraphical profile, this peak correspond to fundamental resonance
frequency ($f_0$) of the site (Tokimatsu, 1997; Bard, 1999; Bonnefoy-Claudet et al., 2006). The averaged
horizontal component of H/V curve contain the contribution from both Rayleigh and Love wave and
some fraction of body waves as well. In a joint inversion of H/V curve with dispersion curve this other
elastic wave effect presence in the H/V curve might bias the retrieved s-wave velocity profile.
Therefore, the presence of Love and other elastic waves existence in the H/V curve must be assumed or
estimated before the inversion process (Bonnefoy-Claudet et al., 2006).Here in this communication we
will try to list the different approaches used for the refining of H/V curve by removing unwanted
fraction (Love wave effect presence)  prior to the  joint inversion with dispersion curve. At present there
are two main research lines describing the H/V curve by taking in account the whole ambient-vibration
wave field, and another just studies the surface wave and Rayleigh wave dominancy in noise wave field
(Lunedei & Malischewsky 2015). Sánchez-Sesma et al (2011) proposed that seismic noise field can be
consider as diffusion-like situation which contain all type of elastic wave (surface and body waves) . He
suggested that the average autocorrelation of the motions at a given receiver, in the frequency domain,
measures average energy density  (DED) and is proportional to the imaginary part of the Green function
(GF) when both source and receiver are the same. The surface wave dominancy opinion of noise wave



field is in favor of Rayleigh wave dominancy (Yamamoto, 2000  Boremann 2002, Cornou, 2002 ,
Okada ,2003). We will try to check both these assumptions here with the borehole data.
The site for which this analysis are made is a borehole site at university of Sao Paulo shown in Fig.1.
The noise measurement  were made with broadband 3 component seismometer nanometrics Trillium
Compact 120-s. Ambient noise wave field measurements were  made for 24 hours on weekend night to
minimized cultural noise  influence. The data acquisition of ambient noise has been done following the
guidelines developed under the SESAME (2004) recommendations. To obtain the fundamental
frequency of the site a window of one-hour recording were processed and the reliability condition
proposed by SESAME (2004) for the H/V curve and peak were followed.

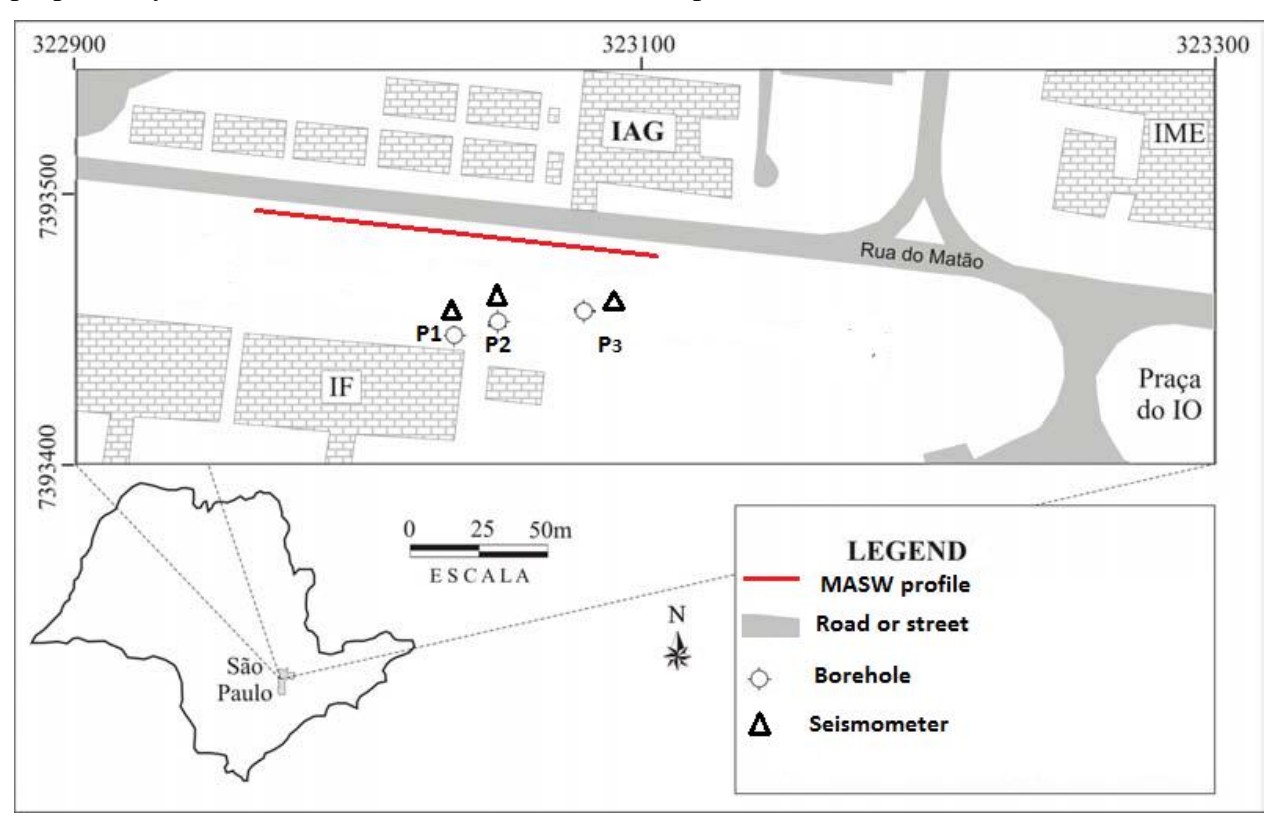

Fig. 1 shows the location map studied area ,legends explain the different symbols. (modified from Porsani 2004).
We will try to discuss briefly here and analyzed our seismic noise data with this diffuse-field
assumption (DFA). Later we will analyzed our data with assumption that H/V curve basically reflect the
Rayleigh wave ellipticity.

**2. Diffuse field assumption technique.**




Sánchez-Sesma et al (2011) proposed to considered ambient noise wave field as diffuse wave field
which contain all different types of waves (surface and body). The ambient noise wave field is
generated by multiple random uncorrelated forces/sources near to or at the earth surface. The  wave
field may contain the scattering effect of various elastic mode. The field intensities could be in a better
way described by diffuse like situation. To assume that the noise wave field is diffuse, the H/V curve
can be estimated for a receiver at earth surface in term of green tensor imaginary part at the source
(source and receiver are assumed to be at same location). The work of Sánchez-Sesma provides an idea
of  linkage between energy density and imaginary part of GF in 3D (energy densities of the noise wave
field is proportional to the  imaginary part of green tensor). The H/V curve obtained from the square
root ratio of  imaginary parts of GF (horizontal and vertical components) Eq.2 serve as intrinsic
property of medium therefore its inversion can be used to retrieved subsurface soil profile.  The detailed
analysis of the method is beyond the scope of this communiqué, interested readers are referred to
Sánchez-Sesma et al (2011) for detailed procedure. The summary of this procedure is that
autocorrelation of motion at a receiver sensor in a given direction is proportional to directional energy
density (DED),and this DED is proportional to the imaginary part of Green tensor at that sensor
(Sánchez-Sesma et al 2011).  Patron et al,( 2009) showed that in case of 3D homogeneous elastic half
space, the horizontal displacement (radial and transverse) have fix energy proportion (e.g $E_1(x,x,\omega) =$
$E_2(x,x,\omega)$ and also $ImG_{11}(x,x,\omega) = ImG_{22}(x,x,\omega)$). For a diffused wave field the H/V can be
represented in term of directional energy densities  assuming source and receiver lies at same location
$(x)$ on the surface of earth as

$\frac{H}{V}(\omega) = \sqrt{\left(\frac{E_1(x,x,\omega)+E_2(x,x,\omega)}{E_3(x,x,\omega)}\right)}$  (1)

$\frac{H}{V}(\omega) = \sqrt{\left(\frac{ImG_{11}(x,x,\omega)+ImG_{22}(x,x,\omega)}{ImG_{33}(x,x,\omega)}\right)}$  (2)
where in (1)

$E_m(x,\omega) = \rho\omega^2\langle u_m(x,\omega)u_m^*(x,\omega)\rangle$   where m = 1,2,3


$= -2\pi\mu E_s k_s^{-1} Im[G_{mm}(x,x,\omega)]$





where energy density is find out at point x in direction m. $\omega, \rho \; and \; u_m$ are angular frequency , layer
density and displacement at point $x$ respectively . $E_s = \rho \omega^2 s^2$ is the strength of diffuse illumination in
term of shear wave average energy density  $\mu$ is shear wave modulus $\langle ... \rangle$ bracket shows the azimuthal
average, $k_s = \frac{\omega}{V_s}$ shear wave number $V_s$ shows medium S-wave velocity. The symbol $(.^*)$ show complex
conjugate process, the medium response in a direction m (of impulse load and acting in same direction)
is indicated by $G_{mm}$ .The H/V curve obtained in this manner are linked to the intrinsic property of
medium , The resulted H/V curve from the  diffuse-field approach might allow its inversion without
considering any supplemented information (dispersion curve).

For our analysis, the data of seismic noise recorded at borehole test site were analyzed with DFA
(diffuse field assumption) frame work of Eq.2 , $\frac{H}{V}(\omega)$ result are obtained from the data with an
integration step of 1000 and window length 40s.The curve obtained by this directional energy density
approach is compared with borehole ellipticity Fig.2. The peak and trough of H/V curve obtained
(Fig.2) correspond to peak and trough of ellipticity. However the shape of H/V curve is generally higher
except the peak ,it is   because of other elastic wave phases contribution. This peak and trough
correlation of the H/V curve with borehole ellipticity shows that at these singularities ( peak and trough)
Rayleigh wave contribution dominate the wave-field. It is important to note here the inversion of
ellipticity curve with dispersion curve are recommended around the peak  region of H/V curve (Picozi
et al,2005 Hobiger et al, 2013). In next section we will focus our attention on the other line of research
which is in the opinion of,  that noise wave field is dominated by surface wave especially Rayleigh
wave ( see section 3) , and H/V curve can be explained by its correlation with Rayleigh wave ellipticity
curve (Bard 1999).



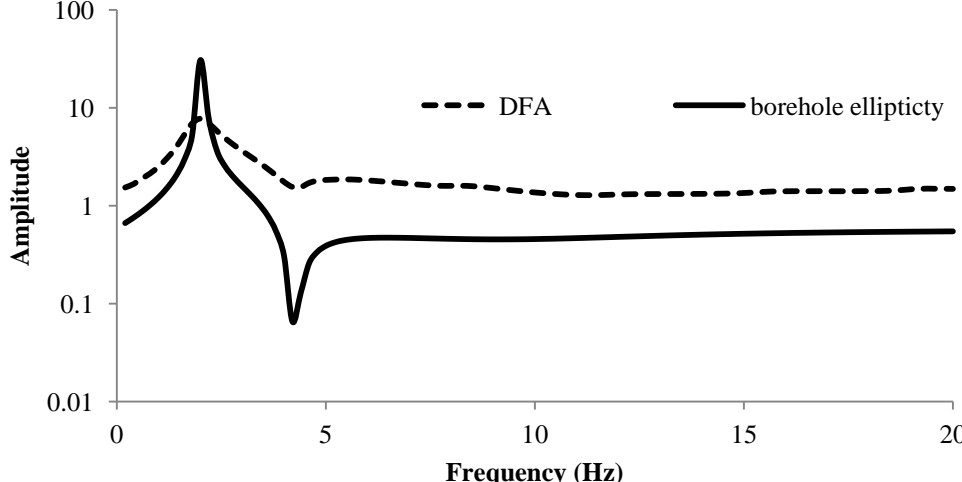



Fig.2 H/V curve obtained through DFA technique at the test site (IAG-USP), the Borehole ellipticity curve is plotted in solid
for comparison at the site.

**3. Surface wave dominancy of seismic noise wave field.**


To find out that whether body or surface waves dominate the noise wave field is analyzed by Bormann (2002) , he used sensors for earthquake and seismic noise recording both at the surface and in the boreholes at different depth levels and concluded the surface-wave nature of seismic noise Fig. 3. Bormann (2002) showed that, the penetration depth of surface waves increases with wavelength, high frequency noise attenuates more rapidly with depth. In case of Fig.3 the noise power at 300 m depth in a borehole was reduced, as compared to the surface, by about 10 dB, at f = 0.5 Hz, 20 dB at 1 Hz and 35 dB at 10 Hz. This continuous  amplitude decline with frequency is in accord with the surface waves nature of seismic noise.




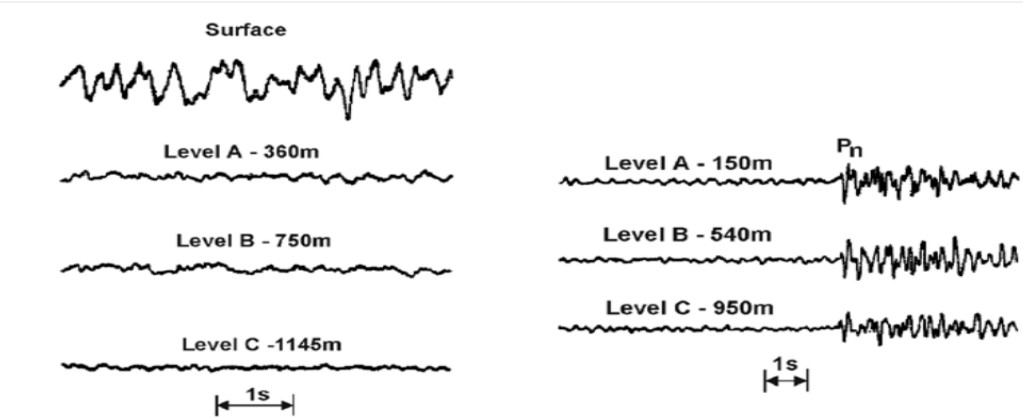


Fig.3 Recording of seismic noise (left) and earthquake signals (right) at the surface and at different depth levels of a
borehole. (Bormann 2002).

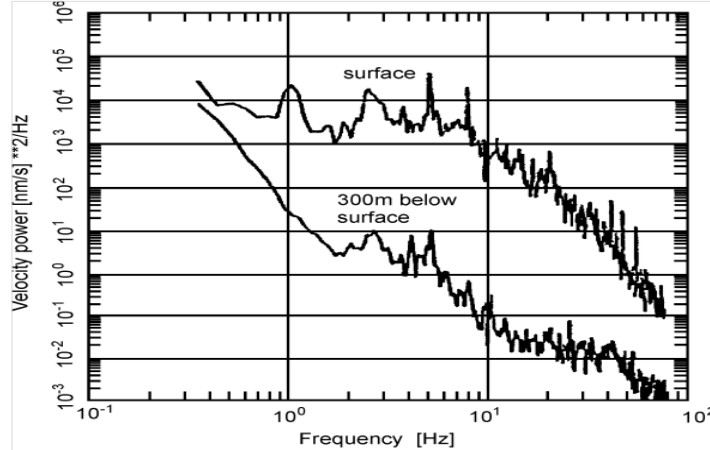


Fig. 4 Velocity power density spectra as obtained for noise records at the surface (top) and at 300 m depth in a borehole
(below) near Gorleben, Germany (Bormann , 2002).
If the noise wave field is mainly dominated by surface wave then another question arise , that what is
the fraction contribution  of Rayleigh and Love waves to the noise wave field. Most of the researchers
focused their attention to find the fraction Rayleigh to Love ratio from the analysis of noise wave field
recorded on vertical component (Li et al., 1984; Horike, 1985; Yamanaka et al., 1994). The results of
these studies showed an agreement in one aspect that microseism (<1Hz) are mainly dominated by
Rayleigh waves however at high frequency ( > 1Hz) a combination of P and Rayleigh wave exists

Table 1 summarize the result of  previous studies on this issue.





|  | Rayleigh waves(%) | Love waves(%) | frequency range(%) |
|---|---|---|---|
| **Chouet et al.,1998** | **30%** | **70%** | **>2Hz** |
| **Yamamoto, 2000** | **<50%** | **>50%** | **3-10 Hz** |
| **Arai et al., 1998** | **30%** | **70%** | **1-12 Hz** |
| **Cornou, 2002** | **60%** | **40%** | **< 1 Hz** |
| **Okada (2003)** | **<50%** | **>=50%** | **0.4-1 HZ** |
| **Köhler(2006)** | **10–35%** | **65–90%** | **0.5–1.3 Hz** |


Table.1 Summary conclusions about the proportion of Rayleigh and Love waves in noise, after different authors (from
Chouet et al., 1998; Yamamoto, 2000; Arai et al. 1998; Cornou 2002, Okada 2003, Köhler2006 ).
**4.Removal of Love wave from horizontal component.**
Rayleigh wave are formed by the linear  pairing of P (primary waves ) and Sv (vertically polarized
shear waves) waves (Aki, 2002).This pairing of vertical and horizontal components have a phase shift
of $\pm \frac{\pi}{2}$, the particle motion induced by Rayleigh wave will dipict an ellipse, this elliptical motion will
either be retrograde or prograde depending on the sign of phase shift. Similarly Love wave is composed
of horizontally polarized shear waves (Sh). The horizontal over vertical axes of ellipse described by
particle motion under the Rayleigh wave influence is term as ellipticity. At situation of homogeneous
half-space the particle motion is retrograde at all frequencies and ellipticity is constant . However in
case of layered structure ellipticity exhibit a peak and trough and the particle motion switch from
retrograde to prograde and then to retrograde with the frequency,  depending on the velocity contrast
between the soil and bedrock ( Konno and Ohmachi, 1998).



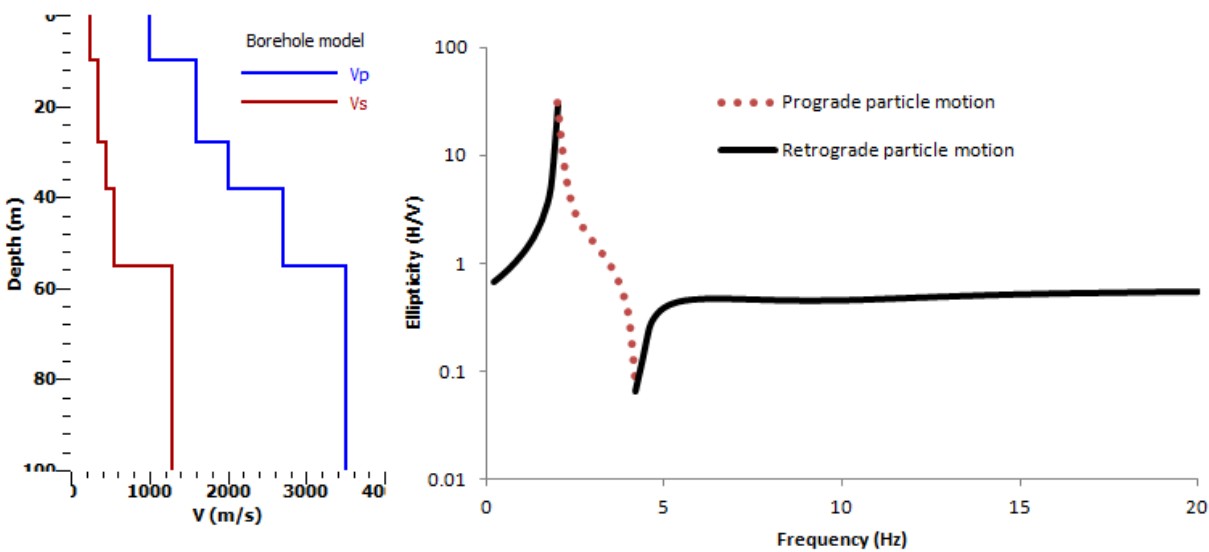



Fig.5 Borehole model at IAG-USP . The ellipticty curve of fundamental Rayleigh wave of the borehole model , marking the frequency ranges where the retrograde and prograde motion might occur.

The conclusion from the preceding section can be drawn that, the contribution of Love wave to the horizontal is not predictable and fluctuate with frequency and from site to site. The  H/V are linked to the ellipticity of Rayleigh wave , in situation where the high shear wave contrast exist between soil and bedrock (Bard,1999).The H/V curve corresponds nicely to the peak of ellipticity curve (Fig.6). The deviation between curves can be easily linked to the presence of Love wave contribution to noise wave field at the horizontal component. We will try to review all the available technique for the this task and compared its result with borehole ellipticity.

H/V curves are obtained and compared with the elliptiity curve of borehole at the same site. The deviation between the curves are due to Love wave contribution. H/V curve is generally higher than the ellipticity curve except at peak frequency of Rayleigh wave ellipticity Fig.6. Three different polarization technique are used to minimized the effect of Love wave. The first technique is the simple H/V of seismic noise (Fig.6)  , In this technique the polarization mean the division of Fourier spectral amplitude of averaged horizontal component over vertical component with the assumption that around the fundamental resonance frequency the vertical component is dominated by Rayleigh only. Generally it is



believed that  Rayleigh (P-Sv) and Love wave (Sh) contribute equally to the horizontal component. Fäh
et al (2001) proposed the division of H/V spectral amplitude by  $\sqrt{2}$ for Love wave effect minimization
from horizontal component Fig.6. However this is not a wise approach because the energy partition of
horizontal component between Rayleigh and Love is not constant and varies with frequency and site
(Köhler *et al.* 2006; Endrun 2011) Table 1. There are two other polarization approaches employed for
this job recently are: time-frequency analysis (Fah et,al 2009)   and RayDec (Hobiger et, al, 2009)
followed by a concise introduction , the interested readers are referred for detail of these approaches
(Fah et,al 2009) and (Hobiger 2009).

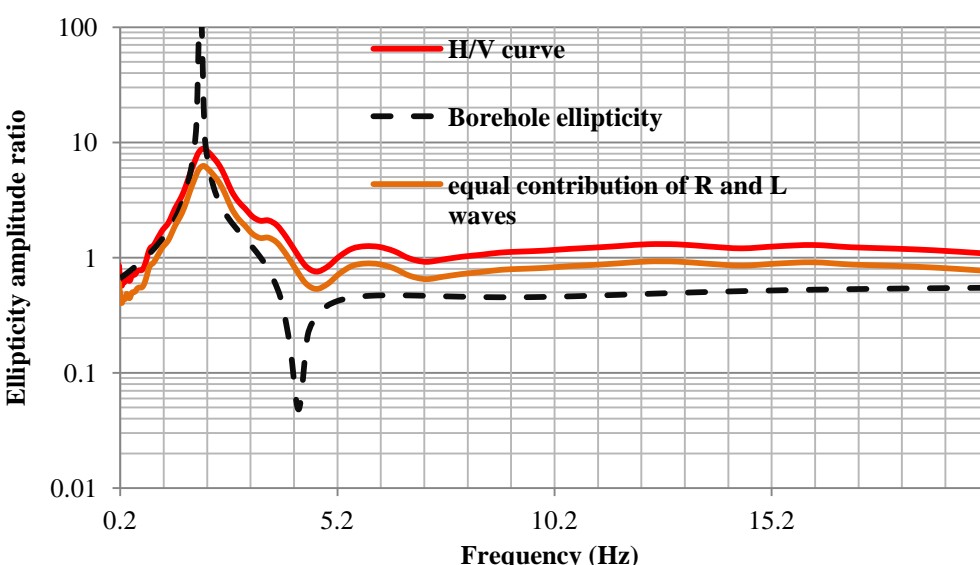


Fig.6 shows the comparison of H/V curve of experimental data recorded at test borehole site at university of Sao Paulo.

**5. Time-frequency analysis.**

In time-frequency analysis the vertical component of noise wave field are considered as a trigger and its
energy level are estimated at given time and frequency and are correlated with horizontal components
(east-west , and north- south). A brief description of the technique is that continuous wavelet
transformation (CWT) are performed on the eastern e(t) , northern n(t) and vertical v(t) components of
noise wave field. CWT (continuous wavelet  transform) transform a signal to time-scale plane. The
scale is a single parameter which controls both the duration and bandwidth.CWT is defined as.





$CWT_{x(t)}(a, b) = \frac{1}{\sqrt{a}} \int_{-\infty}^{\infty} x(t)\psi^* \left(\frac{a-b}{a}\right) dt$        (3)
where x(t) is real-valued signal component {where x(t) = e(t) or n(t) or v(t) }, *ψ(t) shows the mother*
*wavelet,*\* shows a complex conjugation process, a scale parameter (scale control both the duration and
bandwidth) b is translation in time. Fah et,al ( 2009) used a modified Morelet wavelet in a code written
for this job due the reason that traditional Morelet wavelet does not act well for H/V analysis. The
modified Morelet transform is defined as
$\psi(t) = \exp{(i2\pi ft)}e^{\left(\frac{-t^2}{m}\right)}$        (4)
here f is the central frequency of the wavelet, m control the resolution of both frequency and time, low
m mean more time localization on the expense of frequency resolution and higher m result on the
contrary i.e increase frequency resolution at the expense of time resolution. The classical Morelet
wavelet is achieved when m=1/2. After the application of CWT to each component of a single 3-c
sensor recording { e(t), n(t) or v(t) } give rise to three signals amplitude which is both the function of
time and frequency. Two horizontal (e(t), n(t) ) component are merged together to a single component.
As Rayleigh wave are the result of the coupling of p and vertically polarized S-waves, The vertical
component is sorted out for amplitude maxima at each frequency and time translation . Similarly, the
horizontal component is analyzed for amplitude maximum at given frequency and time, the horizontal
components are phase shifted ± at a quarter of the period. This shift of period is done because of
theoretical phase shift between $\pm\frac{\pi}{2}$ vertical and horizontal component particle motion. For each
maximum on the vertical component, the corresponding maximum on the horizontal component are
stored and the ratio of H/V are estimated. There is only one tuning parameter m ,the effect of its
different values of m are shown in Fig.8 , Also the effect on the length of the recorded signal is shown
in Fig.7. This whole process is statistically analyzed by histogram for all the frequency and translation
of time and H/V is obtained by the maximum of each histogram. For detail theoretical outline of this
analysis please read (Fah et,al, 2009).




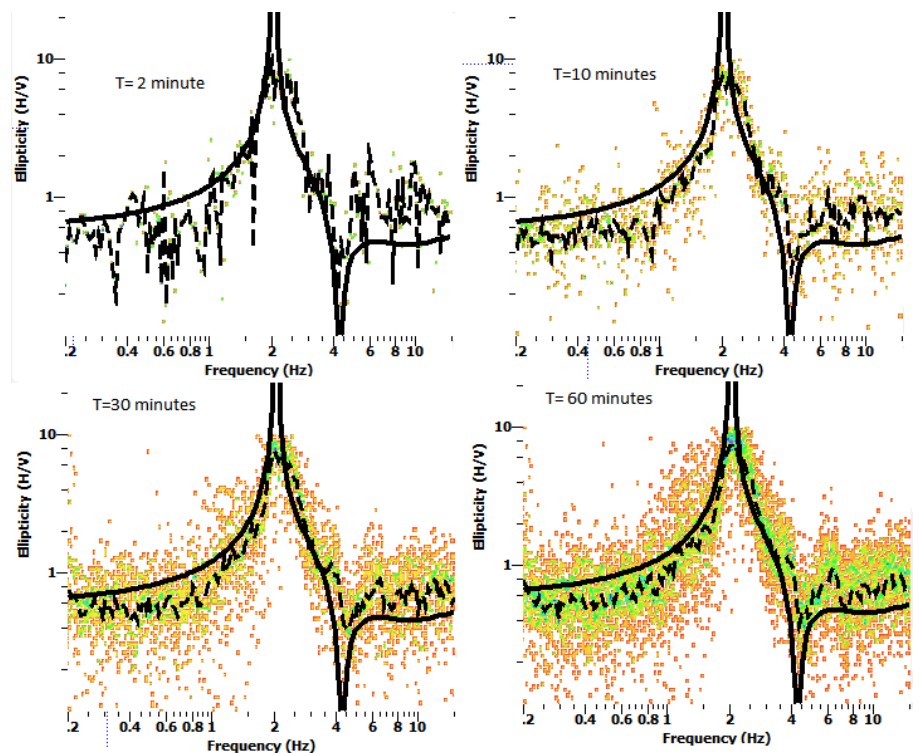


Fig.7 Ellipticity (H/V) obtained from continuous wavelet transformation CWT for different length of the recorded signal; a

histogram is drawn for each frequency, the color within histogram indicates the energy level. (dashed line shows ellipticity

obtained from CWT while solid line shows the ellipticity curve obtained from borehole data at same location - IAG-front)


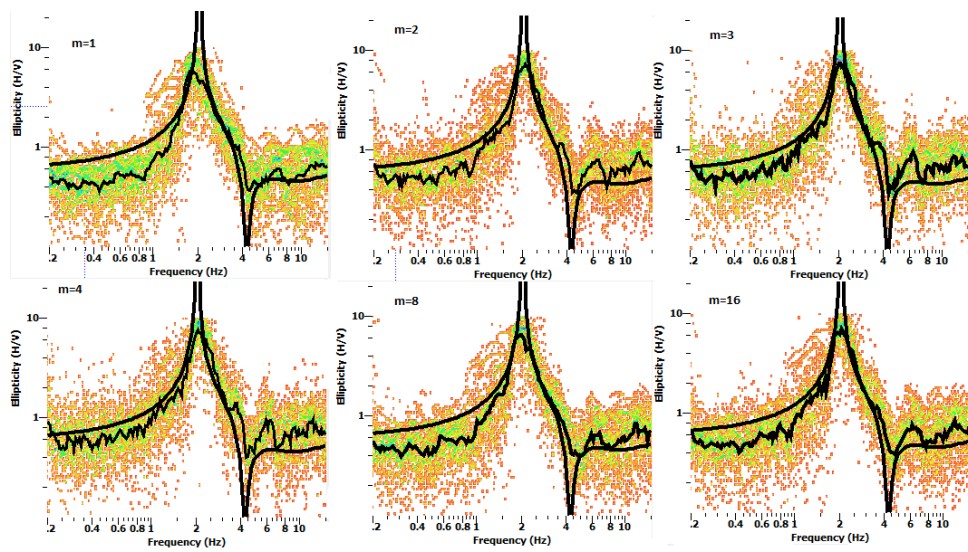







Fig.8 Ellipticity (H/V) obtained from continuous wavelet transformation CWT for different value of m; (curvy line shows
ellipticity obtained from CWT while solid line shows the ellipticity curve obtained from borehole data at same location -
IAG-front)

Fig.7 and 8 show a better result when compared with the borehole ellipticity curve of borehole. The
result of wavelet transform Fig.7&8 shows that the  ellipticity of borehole is retrieved in better way(
especially at right limb of the ellipticity) when the recorded length of the signal is greater than 30
minute ,and when the value of m ≥ 8.

**6 Random decrement technique (RAYDEC)**.

Another polarization technique use for the effect of Love (Sh) wave effect minimization for a single 3
component sensor recording is term as RayDec technique (Hobiger et.al, 2009) The vertical component
is taken as master trigger as because the Sv arrival occur only on the vertical component, while stacking
a large number of horizontal component to obtained ellipticity of Rayleigh curve (Hobiger  et al., 2009)
showed that the obtained ellipticity will be closer to the true ellipticity curve rather than the H/V curve.
To elaborate RayDec techniques consider a  signal {where x(t) = e(t) or n(t) or v(t) }. These three
time series has N number of data points and having length T. To obtain a Rayleigh wave ellipticity
curve, the main idea of  this method is to obtain only Rayleigh waves in comparison of other waves type
by the addition of a large number of filtered signal windows Δ, estimates the energy level of horizontal
averaged and vertical signal in a location where the vertical component change its sigh from -ve to +ve.
Due to the phase shift of $\frac{\pi}{2}$ between Rayleigh wave vertical and horizontal component, both the
horizontal component of EW and NS are projected by factor ϕ with north direction in such a way to
maximized the correlation between the summed horizontal and vertical component. The Rayleigh wave
ellipticity is obtained latterly from Eq.5 . The ellipticity  E is calculated as the square root of the ratio of
the energies in the signal window. Δ.
$$E = \sqrt{\frac{\int_0^\Delta hf^2 . s(t) * dt,}{\int_0^\Delta vf^2 . s(t) * dt}} \qquad\qquad (5)$$



*where* hf.s (t) is the horizontal average ( NS north-south , EW east-west) signal and Vf.s (t) is the
vertical component. This process is repeated for a whole record for an increment of  window length $\Delta$
.The window length is taken as function of frequency such that to ensure 10 significant cycle at chosen
frequency ($\Delta$=10/f where f is the analyzing frequency from 0.2 to 20 Hz are used). There are two tuning
parameter for this technique , $\Delta$ (window length) and df (width of frequency filter). The effect of these
two tuning parameter on the result  is shown Fig.9. For our analysis we used one-hour records of noise,
which were divided into 6 segments of 10 minutes each , afterward, the result is average out for all the
six segments with the aim for minimization of misfit (see for detail procedure Hobiger et al., 2009). The
result of ellipticity obtained via RayDec show a better match with borehole curve especially at right and
left limb.
The analysis of both time-frequency and RayDec for H/V give satisfactory result for the ellipticity
retrieval of Rayleigh waves. Fig 10 shows the comparison of all the technique retrieved curves with that
of borehole ellipticity.




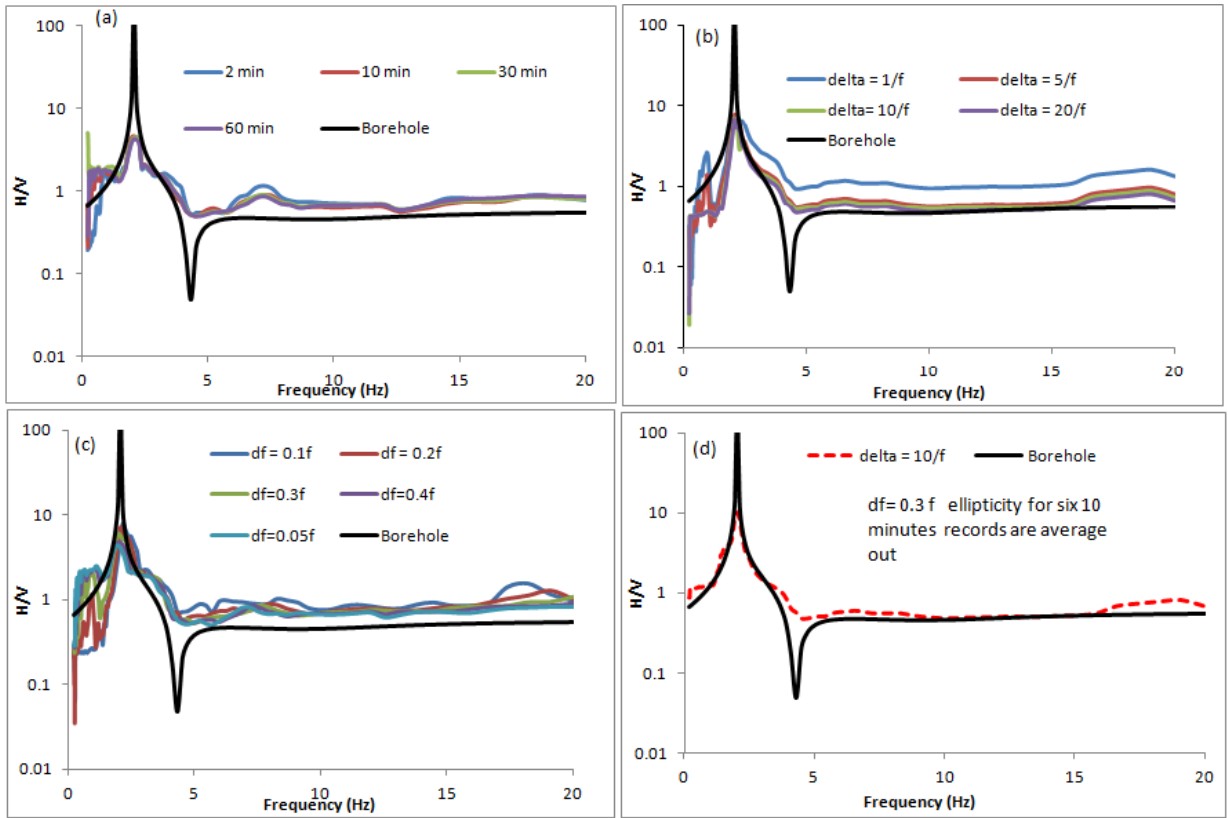


Fig.9 Ellipticity obtained via RADEC technique , (a) shows the effect signal duration on the result.(b) shows the effect of
using different width of window (c) shows the effect of different filter width on the result (d) shows  six  10-minute windows
(ellipticity for each 10 minute is obtained at the end the result is average and compared with borehole data black line).





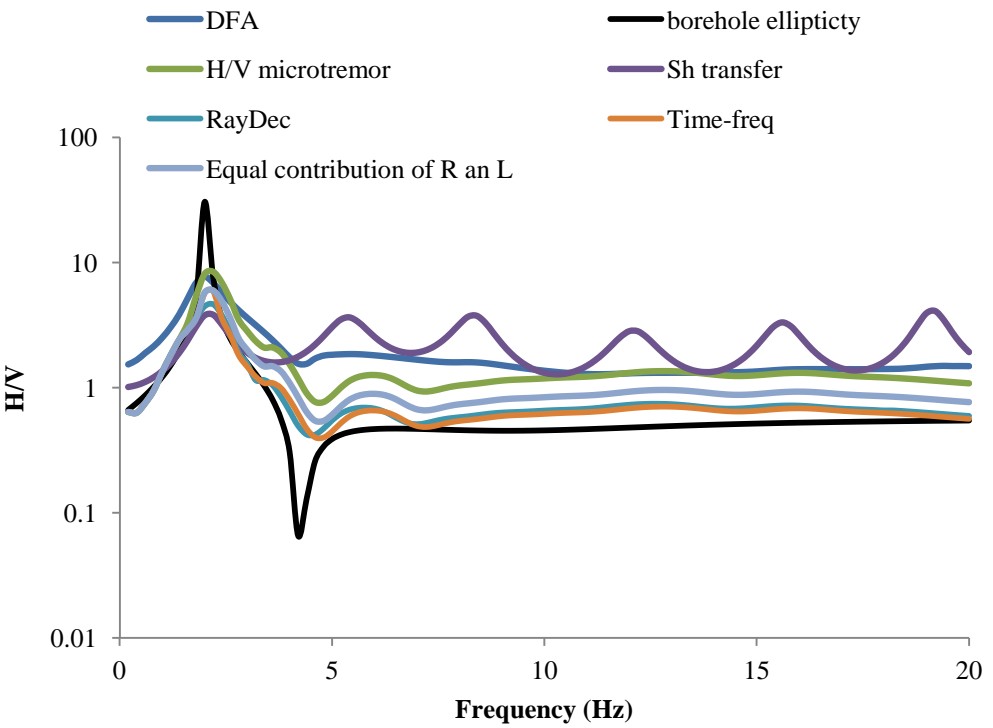

285          Fig.10 Show the comparison of borehole ellipticity curve combine with that of all elliptcity retrieval technique.

**7.Discussion and conclusion.**
Deep soil shear wave velocity information can be retrieved from the joint inversion of H/V curve with
dispersion curve. The H/V curve are assumed to be linked with Rayleigh wave ellipticity for a site.
However in real situation the H/V curve technique can not completely replicate the Rayleigh wave
ellipticity. It is because of the presence of different elastic wave influence presence in the H/V curve
retrieval. Diffuse field approach is a better way to consider the effect of all elastic seismic phases (body
and surface wave). However the analysis of our data shows that at singularities (peak and trough)
especially at peak of the H/V curve obtained via DFA is very well correlated with Rayleigh ellipticity
curve in term of these singularities frequency. This matching can be assumed that around the peak
frequency even in case of diffuse like field situation the seismic noise field is dominated surface wave
especially Rayleigh waves.
The other approach used for the H/V curve technique is linked to the surface wave dominancy
especially Rayleigh wave around the singularities. Three different polarization technique for retrieval of





ellipticity curve by minimizing the Sh contribution to the horizontal component are simple H/V with
equal contribution of Rayleigh and Love wave, time-frequency analysis and RayDec technique. The
result of equal assumption of Rayleigh/Love wave fraction is unable to match with parts of borehole
ellipticity except at peak. However ellipticity curve retrieved from noise analysis through time-
frequency and RayDec shows a good replication of borehole ellipticity around the peak, right and left
limb.
For the deep soil deposit the joint inversion of ellipticity and dispersion curve of Rayleigh wave are
recommended. However due to the presence of effect of other seismic elastic phases especially Love
wave may produced some bias result. Therefore it is extremely necessary to retrieved a H/V curve
where the effect of Love wave contribution are minimized prior to the inversion with dispersion curve.
For our test site we found that time-frequency and RayDec show better result ,by replication left and
right limb of ellipticity curve of the borehole at the site.
**Acknowledgement.**
We are thankful to the Sanchez-Sesma (Instituto de Ingeniería Universidad Nacional Autónoma de
México) for sharing his code for DFA analysis of seismic noise. We appreciate the help from of
Hobiger (ETH Zurich) and Marc Wathelet (University Joseph Fourier - Grenoble 1, Grenoble) . We are
indebted to Marcelo Assumpcao,  Galiardo , Felipe Neves (Instituto de Astronomia, Geofísica e
Ciências Atmosféricas, university of Sao Paulo) for providing the instruments  for recoding and
technical assistance. This work is completed with help of TWAS-CNPq for fellowship grant number
190038/2012-8 (CNPq/TWAS - Full-Time Ph.D. Fellowship - GD 2012 ) for financial support.

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
