# Peer review of "The analysis of H/V curve from different ellipticity retrieval technique for a single 3c-station recording."

_Natural Hazards and Earth System Sciences, 2016_

## Referee Comment (RC1) · Anonymous Referee #1 · 26 Dec 2016

The authors use different approaches from the literature for studying the contribution of the Rayleigh waves to the H/V curve, comparing the obtained results with the expected ellipticity curve of a well-known site.

They do not contribute with any different approach or advance in the study of the wave composition of the noise signal.

They have used only one site for the study. Thus, they compare the obtained results with the ones known from a borehole. A more robust work would need the analysis of different sites with different soil characteristics. Besides, the characteristics of the site (lithology or Vs profile is not provided)

[Figure]

As they comment in the introduction section, they "... try to list the different approaches used for the refining of H/V curve by removing the unwanted fraction (Love wave effect presence) prior to the joint inversion with dispersion curve". But, how do the different studied approaches influence in the estimated soil profile ? How do some of the small observed differences modify the Vs profile? It has been not analyzed.

The English writing is really very poor.

---

## Author Comment (AC1) · 26 Dec 2016

Thank you Sir, for your time and comments. The idea of the work is to check the available different techniques for modeling the H/V curve. As discussed in the text a recent advancement of H/V modeling is suggested by Sánchez-Sesma et al (2011). Who established that one's might invert the H/V curve alone for soil structure Vs profile directly because of the fact that the H/V are linked to Green function, and should be considered as the intrinsic property of the soil. However very recently ( 5th November, 2016) an article appeared online in the Geophysical Journal International by the Jose Pina Flores, suggested the joint inversion of H/V curve with dispersion curve because of non-uniqueness of the problem (though the H/V curve obtained here is achieved

by the Green function analysis).The point I want to make is that If the Green function based modeling of H/V is not capable of Vs retrieval by using the H/V curve alone and it is necessary to include dispersion curve to reduce the non-uniqueness for inversion. In such situation We believe that Rayleigh wave ellipticity approach of H/V modeling is a realistic one because both dispersion and ellipticity curve are attributed to the same wave type (Rayleigh wave) not like The DFA approach which considered all wave type (body and surface waves) contribution for the H/V curve and later its inversion with dispersion curve (which is based on Rayleigh wave only)

**The surface wave as whole and the Rayleigh wave in particular dominated in seismic wave filed, is demonstrated by some of the previous studies perform for noise wave field characterization. We tried to check if we have a control site (borehole) where we could have noise recording and check these different, modeling approach for H/V curve we will be able to see which modeling analysis give a better match with experimental data.**

**The Vs profile is given in Fig.5. However I will give the borehole model below as well. Borehole model. Thickness (m) Vp(m/s) Vs (m/s) Density (kg/m3) 10 1000 270 1700 18 1600 360 1700 10 2000 455 1700 17 2700 550 1700 bedrock 3500 1300 2000**

**Yes the analysis is done for one site because of the unavailability of borehole Vs profiles at our disposal, for the other sites ( the Vs measurement were not done at the time of well-logging of the other boreholes), the analysis is done therefore for this site because of Vs measurements availability, which is required to obtain the theoretical ellipticity for the site by forward modeling from the obtained model and compared it with experimental data.**

**Regarding the sensitivity analysis of the H/V/ellipticity curve to the soil properties such as Vs,Vp, thickness and density is already done by Wathelet, 2005 and Hobiger, 2013 for the Rayleigh wave ellipticity approach.(As the modeling of the H/V is done with ellipticity of Rayleigh wave, hence the mentioned studies especially the Wathelet**

2005 has checked all the parameter which can affect ellipticity curve)

**About the English I am sorry for it I am trying my best in improving it.**

References mentioned in response.

M. Hobiger,, C. Cornou,1 M. Wathelet,G. Di Giulio,B. Knapmeyer Endrun, F. Renalier,P.-Y. Bard,1 A. Savvaidis, S. Hailemikael,N. Le Bihan,M. Ohrnberger and N. : The Ground structure imaging by inversions of Rayleigh wave ellipticity sensitivity analysis and application to European strong-motion sites Geophys. J. Int. 192, 207–229.2013.

Marc Wathelet - Array recordings of ambient vibrations: surface-wave inversion.PhD thesis, Liège University (Belgium), 2005

Sanchez-Sesma, F.J. : A theory for microtremor H/V spectral ratio: application for a layered medium, 374 Geophys. J. Int., 2011, 186(1), 221–225. Piña-Flores J., Perton M., García-Jerez A., Carmona E., Luzón F., Molina-Villegas J.C., Sánchez-Sesma F.J. (2017). The inversion of spectral ratio H/V in a layered system using the Diffuse Field Assumption (DFA), Geophysical Journal International, In press. [doi:10.1093/gji/ggw416]

Please also note the supplement to this comment:
http://www.nat-hazards-earth-syst-sci-discuss.net/nhess-2016-370/nhess-2016-370-AC1-supplement.pdf

---

## Referee Comment (RC2) · Anonymous Referee #2 · 30 Jan 2017

The paper entitled ""The analysis of H/V curve from different ellipticity retrieval technique for a single 3c-station recording" (by Ullah & Prado), deals with the comparison of H/V spectral ratio from various techniques with the ellipticity curve for a selected pilot borehole site at Sao Paolo, Brasil. The manuscript could be published in the Journal NHESS only after the following remarks/suggestions are taken into account.

General Remarks 1. After the comparison of the H/V spectral ratio from different techniques (time frequency, microtremors, DFA etc.) with the borehole ellipticity, the authors conclude that the time-frequency and RayDec show better results in replicating the left and right part of the borehole ellipticity curve. In turn, they indirectly claim that the joint inversion of H/V spectral ratio curve and dispersion curve of Rayleigh waves

should provide better estimate of the velocity structure of the borehole when using the time-frequency and RayDec techniques. I would suggest to the authors: (a) based on the results of all applied H/V techniques and (b) using the same dispersion curve of the pilot borehole site, to estimate the Vs profile of the borehole for the different H/V curves. Then to compare the various inverted profiles with that provided in Figure 5, in order to show the validity of their conclusion.

2. Several references are provided in the manuscript but do not appear in the References chapter (see in attached .pdf)

Specific Remarks Some orthographic or/and syntax errors are given in the attached .pdf.

Please also note the supplement to this comment:
http://www.nat-hazards-earth-syst-sci-discuss.net/nhess-2016-370/nhess-2016-370-RC2-supplement.pdf

**Supplement:**

[revised manuscript text omitted]

---

## Author Comment (AC2) · 31 Jan 2017

Irfan Ullah and Renato Luiz Prado syed_528@yahoo.com

Thank you very much Sir for your time and for highlighting my mistakes. I have corrected all the error/mistake as suggested in text and missing reference. The Idea regarding, to show some result about the TFA and RayDec curve joint inversion with the borehole model dispersion curve is excellent and I have jointly inverted the TFA and RayDec curve with the borehole dispersion curve. A paragraph is added to the article before discussion and conclusion. Here i am giving the new text and the figure which is added to the article. I am adding the corrected article into the supplement as well. 7. Joint inversion of the ellipticity and dispersion curve. The ellipticity curve retrieved from both the time-frequency analysis and RayDec technique are jointly inverted along with

[Figure]

theoretical dispersion curve of the borehole obtained by forward modeling (using code gpdc, http://www.geopsy.org last accessed 1-29-2017). The frequency range of the dispersion curve is considered above the fundamental frequency of the site 2Hz (in our case) till 45 Hz, below the fundamental frequency of the site dispersion curve is difficult to retrieved for Rayleigh waves ,as the medium filter out all the lower frequencies (Scherbaum et al, 2003). To get the 1D shear wave velocity from dispersion and these H/V curve, the modified neighborhood algorithm (NA) proposed by Wathelet ( 2008) are used. In comparison to linearized inversion procedure, NA is a derivative-free procedure. NA is considered very good inversion strategy because it has the advantage over the other approaches as it utilizes all previous model information to sample the new model (Sambridge 1999). The parameters for inversion are considered as follow: the numbers of layers are considered to be four above the bedrock, P-wave velocity are linked with S-wave velocity, S-wave velocity are allowed to linearly increased from surface to the bedrock, density is taken constant at 2000 kg/m3, while the Poisson ratio are considered to change from 0.2 to 0.5. The inversion is made for time-frequency and RayDec based H/V curve only, as that of DFA H/V curve contains the effect of all wave-type and its inversion with Rayleigh wave dispersion curve will certainly give bias result. The misfit between all generated models and target curves (dispersion and H/V) are calculated using eq. 6. misfit=1/N $\sqrt{(\sum((x\_t(f) + x\_m(f))^2 / x\_t(f)^2))} (6) where x\_t(f) is the target curve (either experimental H/V or borehole dispersion curve) at frequency (f)$

**Fig. 1.**

**Supplement:**

**The analysis of H/V curve from different ellipticity retrieval technique for a single 3c-station recording.**

1. Irfan Ullah : Corresponding Author
**Email**: syed_528@yahoo.com
**Telephone** : +5511 3091-2789
**Fax**: +5511 3091-5034
**Adress**: Department of Geophysics - IAG Sao Paulo University - USP Rua does Matão, 1226, Cidade Universitária CEP-05508-090 - Sao Paulo, Brasil.

2. Renato L.PRADO **:** Department of Geophysics - IAG Sao Paulo University - USP Rua does Matão, 1226, Cidade Universitária CEP-05508-090 - Sao Paulo, Brasil.

[revised manuscript text omitted]

$(x,x)$ at frequency $\omega$ and subscript 1,2 show both the horizontal (east-west, and north south)
while 3, that of vertical component. Energy density is find out at point x in direction m.
$\omega, \rho$ $and$ $u_m$ are angular frequency , layer density and displacement at point $x$ respectively
. $E_s = \rho\omega^2 s^2$ is the strength of diffuse illumination in term of shear wave average energy density
$\mu$ is shear wave modulus $\langle...\rangle$ bracket shows the azimuthal average, $k_s = \frac{\omega}{V_s}$ shear wave number $V_s$
shows medium S-wave velocity. The symbol (.$^*$) show complex conjugate process, the medium
response in a direction m (of impulse load and acting in same direction) is indicated by $G_{mm}$ .The
H/V curve obtained in this manner is linked to the intrinsic property of medium , The resulted
H/V curve from the  diffuse-field approach might allow its inversion without considering any
supplemented information (dispersion curve).

For our analysis, the data of seismic noise recorded at borehole test site were analyzed with DFA
(diffuse field assumption) frame work of Eq.2 , $\frac{H}{V}(\omega)$ results are obtained from 50 windows each
of length 50 seconds, each window is normalized as suggested by Jose Pina Plores et al (2016)
.The curve obtained by this directional energy density approach is compared with borehole
ellipticity Fig.2. The peak and trough of H/V curve obtained (Fig.2) correspond to peak and
trough of ellipticity. However the shape of H/V curve is generally higher  except the peak ,it is
because of other elastic wave phases contribution. This peak and trough correlation of the H/V
curve with borehole ellipticity might shows that at these singularities ( peak and trough)
Rayleigh wave ellipticity  dominate the shape of H/V. It is important to note here the inversion of
ellipticity curve with dispersion curve are recommended around the peak  and right limb of the
H/V curve (Picozi et al. 2005 Hobiger et al. 2013). It is to be noted that all this H/V computation
in this case is resting on the assumption that noise wave field is diffused. Is it?.  Mulagia (2012)
statistically check this assumptions of diffused field for noise at 65 different locations for diverse
geological conditions and recording environment. He showed statistically that the seismic noise
wave field is not diffused as the noise wave field is not azimuthally isotropic. In the 
[revised manuscript text omitted]

$\Delta$.

$\mathbf{E} = \sqrt{\frac{\int_0^\Delta \mathbf{hf^2.s(t)*dt,}}{\int_0^\Delta \mathbf{vf^2.s(t)*dt}}}$ (5)

where hf.s (t) is the horizontal average ( NS north-south , EW east-west) signal and Vf.s (t) is the
vertical component. This process is repeated for a whole record for an increment of  window
length $\Delta$ .The window length is taken as function of frequency such that to ensure 10 significant
cycle at chosen frequency ($\Delta$=10/f where f is the analyzing frequency from 0.2 to 20 Hz are
used). There are two tuning parameters for this technique , $\Delta$ (window length) and df (width of
frequency filter). The effect of these two tuning parameter on the result  is shown Fig.8. For our
analysis we used one-hour records of noise, which were divided into 6 segments of 10 minutes
each , afterward, the result is average out for all the six segments with the aim of minimization of misfit (see for detail procedure Hobiger et al., 2009). The result of ellipticity obtained via
RayDec show a better match with borehole curve especially at right and left limb.

The analysis of both time-frequency and RayDec for H/V give satisfactory result for the
ellipticity retrieval of Rayleigh waves. Fig 9 shows the comparison of all the technique retrieved
curves with that of borehole ellipticity.

[Figure]

Fig.8 Ellipticity obtained via RayDec technique , (a) shows the effect signal duration on the result.(b) shows the
effect of using different width of window (c) shows the effect of different filter width on the result (d) shows  six
10-minute windows (ellipticity for each 10 minute is obtained at the end the result is average and compared with
borehole data black line).

[Figure]

Fig.9Show the comparison of borehole ellipticity curve combine with that of all elliptcity retrieval technique.

**7.Joint inversion of the ellipticity and dispersion curve.**

The ellipticity curve retrieved from both the time-frequency analysis and RayDec technique are jointly inverted along with theoretical dispersion curve of the borehole obtained by forward modeling (using code gpdc, http://www.geopsy.org last accessed 1-29-2017). The frequency range of the dispersion curve is considered above the fundamental frequency of the site 2Hz (in our case) till 45 Hz, below the fundamental frequency of the site dispersion curve is difficult to retrieved for Rayleigh waves ,as the medium filter out all the lower frequencies (Scherbaum et al, 2003). To get the 1D shear wave velocity from dispersion and these H/V curve, the modified neighborhood algorithm (NA) proposed by Wathelet ( 2008) are used. In comparison to linearized inversion procedure, NA is a derivative-free procedure. NA is considered very good inversion strategy because it has the advantage over the other approaches as it utilizes all previous model information to sample the new model (Sambridge 1999). The parameters for inversion are considered as follow: the numbers of layers are considered to be four above the bedrock, P-wave velocity are linked with S-wave velocity, S-wave velocity are allowed to linearly increased from surface to the bedrock, density is taken constant at 2000 kg/m3, while the Poisson ratio are considered to change from 0.2 to 0.5. The inversion is made for time-frequency and RayDec based H/V curve only, as that of DFA H/V curve contains the effect of all wave-type and its inversion with Rayleigh wave dispersion curve will certainly give bias result. The misfit between all generated models and target curves (dispersion and H/V) are calculated using eq. 6.

$$misfit = \frac{1}{N}\sqrt{\sum((x_t(f) + x_m(f))^2/x_t(f)^2)}$$ (6)

where $x_t(f)$ is the target curve (either experimental H/V or borehole dispersion curve) at frequency $(f)$
while $x_m(f)$ is the modeled curve for both H/V and dispersion curve, N is the number of points of target
curve considered for inversion. The misfit for both the target ( dispersion and H/V ) are weighted
average. Hobiger et al (2013) analyzed that which part of ellipticity is carrying the most important
information about the surface structure. He concluded that right limb up to the trough is the most
responsive part of curve to the subsurface target. However to constraint the peak of the H/V curve
(which is carrying important  information about the depth of bedrock)  we have considered the left limb
of the H/V curve as well for the inversion , therefore the frequency range of the H/V considered from 1
to 4.2Hz. The result of the inversion are shown in Fig.10. The results shows that in both the cases (Time-
frequency analysis and RayDec based H/V curves),the best fit inverted model  is very near to original
borehole model.

[Figure]

Fig.10 Shows the inversion result of time-frequency Fig.10(a,i) RayDec H/V curve Fig.10(b,i) along with the borehole dispersion curve. Dotted line in both cases (10ab,i,ii) indicate the target curve while soild line show the best fit model response (10ab,i,ii).In Fig.10ab.iii, all the inverted models are shown dashed line borehole model while solid line the bestfit model.

**7.Discussion and conclusion.**

[revised manuscript text omitted]

[22]. Piña-Flores J., Perton M., García-Jerez A., Carmona E., Luzón F., Molina-Villegas J.C., Sánchez-Sesma F.J. (2017). The inversion of spectral ratio H/V in a layered system using the Diffuse Field Assumption (DFA), Geophysical Journal International, In press.

[23]. Porsani , Welitom Borges, Vagner Roberto Elis , Liliana Alcazar Diogo .Investigações geofísicas de superfície e de poço no sítio controlado de geofísica rasa do IAGUSP, Revista Brasileira de Geofísica.**2004**.

[24]. Sambridge, M. (1999). Geophysical inversion with a Neighbourhood algorithm, I, Searching a parameter space. Geophys. J. Int., 138, 479–494

[25]. Sanchez-Sesma, F.J. *et al.*,. A theory for microtremor H/V spectral ratio: application for a layered medium, *Geophys. J. Int.,* **2011**, **186**(1), 221–225.

[26]. Scherbaum, F., K.-G. Hinzen, and M. Ohrnberger,. Determination of shallow shear-wave velocity profiles in Cologne, Germany area using ambient vibrations. Geophys. J. Int.**2003**, 152, 597-612.

[27]. SESAME scientific products(2001-2004) http://SESAME.geopsy.org/SES_Reports.htm

[28]. Tokimatsu, K. (1997). Geotechnical site characterization using surface waves. Earthquake Geotechnical Engineering, Ishihara (ed.), Balkema, Rotterdam, 1333-1368.

[29]. Wathelet, M., (2008). An improved neighborhood algorithm: parameter conditions and dynamic scaling, Geophys. Res. Lett., **35,** L09301,doi:10.1029/2008GL033256.

[30]. Yamamoto, H. (2000). Estimation of shallow S-wave velocity structures from phase velocities of love- and Rayleigh- waves in microtremors. Proceedings of the 12th World Conference on Earthquake Engineering. Auckland, New Zealand, **2000.**

[31]. Yamanaka, H., Takemura, M., Ishida, H, Niwa, M. Characteristics of long-period microtremors and their applicability in exploration of deep sedimentary layers. Bull. Seism. Soc. Am. **1994**., 84, 1831-1841.A.